# Genomic patterns of progression in smoldering multiple myeloma

Niccolò Bolli [1,2,3], Francesco Maura[1,3], Stephane Minvielle[4,5], Dominik Gloznik[3], Raphael Szalat[6], Anthony Fullam [3], Inigo Martincorena[3], Kevin J. Dawson[3], Mehmet Kemal Samur[6], Jorge Zamora[3], Patrick Tarpey[3], Helen Davies[3], Mariateresa Fulciniti[6], Masood A. Shammas[6], Yu Tzu Tai[6], Florence Magrangeas[4,5], Philippe Moreau[4,5], Paolo Corradini[1,2], Kenneth Anderson[6], Ludmil Alexandrov[7], David C. Wedge[8], Herve Avet-Loiseau[9], Peter Campbell[1] & Nikhil Munshi[6,10]

We analyzed whole genomes of unique paired samples from smoldering multiple myeloma (SMM) patients progressing to multiple myeloma (MM). We report that the genomic landscape, including mutational profile and structural rearrangements at the smoldering stage is very similar to MM. Paired sample analysis shows two different patterns of progression: a "static progression model", where the subclonal architecture is retained as the disease progressed to MM suggesting that progression solely reflects the time needed to accumulate a sufficient disease burden; and a "spontaneous evolution model", where a change in the subclonal composition is observed. We also observe that activation-induced cytidine deaminase plays a major role in shaping the mutational landscape of early subclinical phases, while progression is driven by APOBEC cytidine deaminases. These results provide a unique insight into myelomagenesis with potential implications for the definition of smoldering disease and timing of treatment initiation.

[1] Department of Oncology and Hemato-Oncology, University of Milan, Milan, 20122, Italy. [2] Department of Oncology and Hematology, Fondazione IRCCS Istituto Nazionale dei Tumori, Milan, 20133, Italy. [3] Cancer Genome Project, Wellcome Trust Sanger Institute, Hinxton, Cambridgeshire CB10 1SA, UK. [4] CRCINA, INSERM, CNRS, Université de Nantes, Université d'Angers, Nantes, 44035, France. [5] CHU de Nantes, Nantes, 44093, France. [6] Jerome Lipper Multiple Myeloma Center, Dana–Farber Cancer Institute, Harvard Medical School, Boston, 02215 MA, USA. [7] Department of Cellular and Molecular Medicine and Department of Bioengineering and Moores Cancer Center, University of California, San Diego, La Jolla, CA 92093, USA. [8] Big Data Institute, Nuffield Department of Medicine, University of Oxford, Oxford, OX3 7BN, UK. [9] Genomics of Myeloma Laboratory, L'Institut Universitaire du Cancer Oncopole, Toulouse, 31100, France. [10] Veterans Administration Boston Healthcare System, West Roxbury, 02132 MA, USA. These authors contributed equally: Niccolò Bolli, Francesco Maura. Correspondence and requests for materials should be addressed to H.A-L. (email: avet-loiseau.h@chu-toulouse.fr) or to P.C. (email: pc8@sanger.ac.uk) or to N.M. (email: nikhil_munshi@dfci.harvard.edu)

Multiple myeloma (MM) is preceded by a premalignant expansion of clonal plasma cells, recognized as monoclonal gammopathy of undetermined significance (MGUS) or smoldering MM (SMM)[1–3]. While only a small minority of MGUS patients progress to MM, SMM represents a heterogeneous disease where a fraction of patients progresses to symptomatic myeloma rather quickly, and others experience an indolent course. Several clinical features have been identified to stratify the risk of progression of SMM patients[4], and the definition of MM itself has recently been updated to include additional signs of disease burden and organ involvement that predict imminent symptomatic evolution[5]. In fact, there is increasing pressure to reliably identify smoldering patients at high risk of progression, owing to preliminary evidence of survival benefit upon treatment of high-risk SMM[6]. However, genomic markers of disease reflecting the intrinsic biological features of the disease may provide, in the future, a more accurate stratification of SMM than the current clinical and biological criteria, which constitute only a surrogate measure of the disease burden rather than direct measures of its biological features and aggressiveness[4,7–10].

Next-generation sequencing (NGS) has provided a comprehensive characterization of mutations and mutational processes operative on coding regions of the MM genome[11–16]. Much less is known of the genomic features of smoldering stages, where the full genomic spectrum of alterations and its evolution after progression to MM is still largely unexplored. DNA copy number and gene-expression analysis in MGUS and SMM have found similar abnormalities to MM, however they have failed to generate a robust and clinically useful model of progression to MM[4,17–20]. Similarly, preliminary data from limited whole-exome sequencing (WES) approaches have identified some of the recurrent mutations of MM at the premalignant stages[21,22]. However, an important area of interest is to understand the full spectrum of genomic alterations in premalignant stages, including both mutations and structural rearrangements; to identify the molecular processes that generate them; and to explore the patterns of evolution of such abnormalities at progression.

Whole-genome sequencing (WGS) has the potential to provide an unprecedented resolution to interrogate the full repertoire of somatic mutations, copy number alterations (CNAs), genomic rearrangements and mutational processes involved in MM evolution, and has never been performed in asymptomatic MM. In particular, analysis of mutational signatures by WGS can provide information on the mutational processes operative in the cancer cell, and thus shed light on the early pathogenesis of MM[23]. Mutational processes described so far in symptomatic MM using whole-exome data include the spontaneous deamination of methylated cytosines (generating age-related signatures)[23,24] and the aberrant activity of the DNA deaminase APOBEC, an enzyme linked to global and localized hypermutation in a variety of cancers[11,13,16,25–27]. However, contrary to its clear driver role in other postgerminal center lymphoproliferative disorders[28], little is known in MM about the role of activation-induced cytidine deaminase (AID), a DNA deaminase expressed at the activated germinal center B-cell stage, whose canonical signature has only been reported on few specific genes or rearrangements in MM[11]. Since the advent of NGS, NNMF techniques have allowed genome wide analysis of mutational signatures and both the canonical and a second, referred to as noncanonical, AID signature have been identified in chronic lymphocytic leukemia and non-Hodgkin lymphoma but not in MM[23,29,30].

By WGS analysis of unique paired samples, we here provide a comprehensive description of the genomic features of smoldering stages of MM, as well as models of their evolution to MM.

## Results

**The genomic landscape of smoldering myeloma.** We sequenced the whole-genome of 11 SMM cases at diagnosis (Supplementary Table 1). Sequencing was performed to an average depth of 38.7× (Supplementary Table 2), and detected 57,736 somatic base substitutions (range: 2389–7297, median = 5308 per patient) and 4397 small insertion–deletions (indels) (range: 209–541, median = 399 per patient) (Fig. 1a). As expected, most substitutions and indels (98.3%) fell in noncoding regions (Supplementary Figure 1). All patients presented a translocation and/or a CNA described as driver in MM, and all but two showed at least one mutation in a MM driver gene (Fig. 1a)[11,13,31,32]. No recurrent substitutions were observed in noncoding regions. Seven patients were characterized by a hyperdiploid status (Fig. 1a, Supplementary Figure 2); two patients had a t(4;14) and one patient had a t(11;14) translocation (Fig. 1a, b). Strikingly, 5/11 cases showed translocations of MYC, none of which involved the immunoglobulin heavy chain (IGH) locus. In one patient, MYC was translocated with multiple breakpoints suggestive of convergent evolution of independent rearrangement events in different subclones (Fig. 1b). Overall, we observed a median of 35 structural rearrangements per patient, mostly nonrecurrent, in the form of either translocations, inversions, deletions, or internal tandem duplications (Supplementary Figure 3). Analysis of clonal CNAs, i.e., those present in virtually all myeloma cells and thus of early onset, showed frequent trisomies (from hyperdiploid cases) as well as 1q gain and 13q deletions in up to 50% of cases (Fig. 1c). Furthermore, we observed frequent clonal gains in 6p and deletions in 6q and 16q. The integrated genomic landscape of mutations, copy number changes, and rearrangements in our series of smoldering MM thus revealed a much more complex profile than what could be investigated with WES and copy number array data[17,18,21,22]. This is more analogous to MM at diagnosis than of an indolent condition, and suggests that driver events identified in MM are already operative in the earlier stages of the plasma cell disorder.

**Progression to multiple myeloma.** All our patients evolved to symptomatic MM with a median time to progression of 8 months (range: 2–41 months), independent of risk stratification criteria based on serum M-protein concentration, extent of bone marrow involvement or cytogenetic features (Supplementary Table 1). For 10 of 11 patients we could analyze a paired tumor sample to explore the evolution of the genomic landscape at the time of progression. The number of substitutions and indels in symptomatic MM increased in all but one patient (Fig. 2a, Supplementary Figures 4, 5), but was overall not significantly different. However, a relevant fraction of shared mutations significantly shifted their cancer cell fraction (CCF), while others were lost or acquired, suggesting the existence of a dynamic competition between subclones during disease progression. This was analyzed by a hierarchical bayesian Dirichlet process[11] to group mutations with similar CCF into clusters that reflect the subclonal structure of the tumor. Surprisingly, all samples presented one or more clusters of subclonal variants, reflecting the presence of heterogeneity with evidence of continued spontaneous evolution of the disease even in early, premalignant phases (Supplementary Table 3). Furthermore, analysis of paired samples showed two general patterns of evolution to symptomatic MM (Supplementary Figure 6). In 6 of 10 patients, the subclonal composition changed during the evolution from SMM to MM in a branching pattern. This reflects a spontaneous evolution model where, without any external selective pressure from treatment, acquisition of new genetic lesion(s) conferred a proliferative

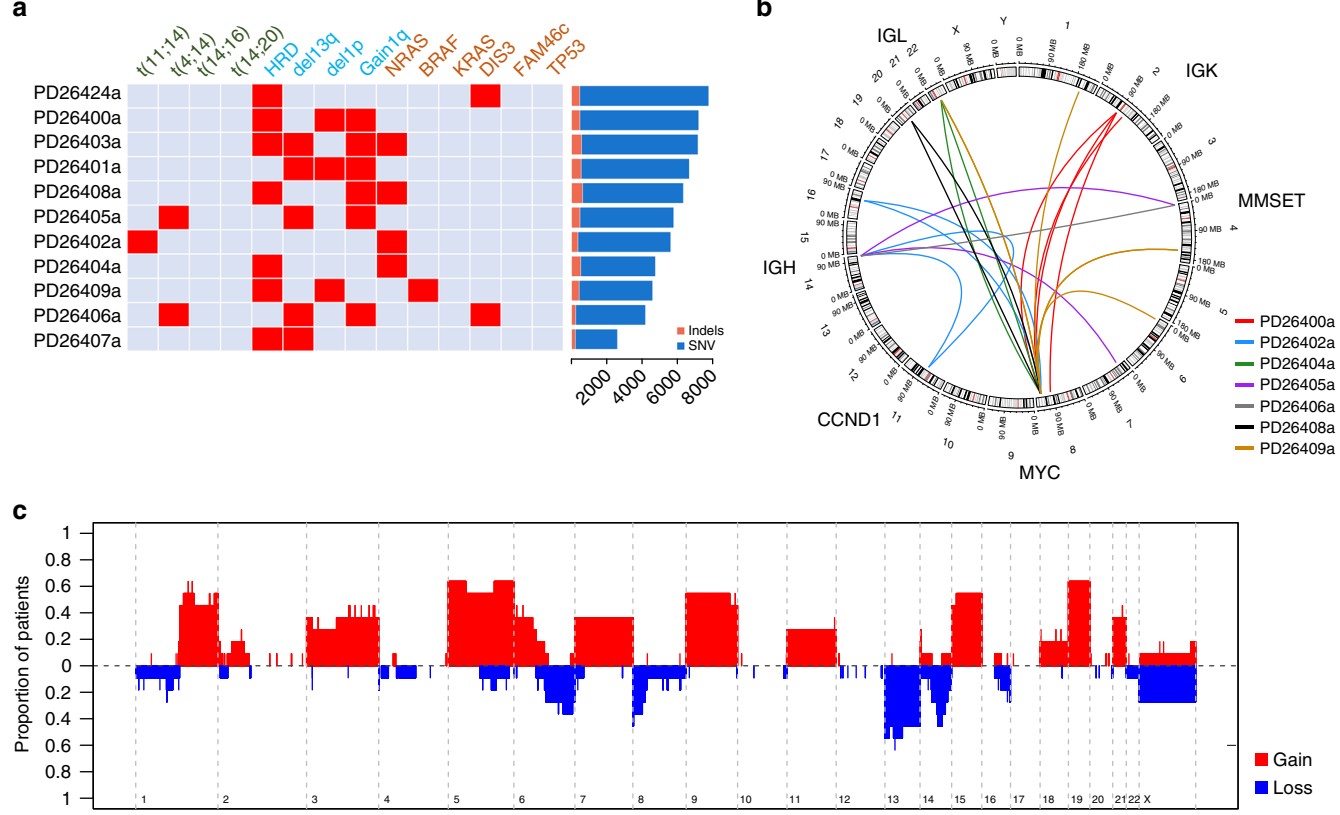

**Fig. 1** The genomic landscape of smoldering multiple myeloma. **a** Map representing the prevalence of known driver events among 11 smoldering MM patients. On the right the bar plot shows the number of somatic mutations (substitutions and indels) in each patient. **b** Circos plot representing the recurrent MM translocations in MM identified in this study (*IGH* and *MYC* genes). **c** Cumulative prevalence of clonal copy number changes

advantage to a subclone at the expense of others (exemplified in Fig. 2b). In the other four patients, all subclones were equally represented in both SMM and MM samples, without any significant change in their subclonal structure (exemplified in Fig. 2c). Known driver mutations could be found in the cluster of clonal mutations, suggesting they can represent early lesions in MM pathogenesis, but also in new subclones acquired during progression (Fig. 2b, c and Supplementary Figure 6). Patients progressing with branching evolution were generally characterized by a longer time to MM progression (median = 23 months; range: 2–41 months), suggesting that acquisition of additional genomic lesions was needed to change the biology of the tumor into a more aggressive phenotype (Fig. 2d). However, one patient progressed in only 2 months with a branching pattern consistent with differential clonal expansion, likely indicating that the first sampling was performed just after the tumor acquired the final clonal sweep that changed the phenotype from a smoldering to an active disease, and just before the new "active subclone" accumulated enough burden to become symptomatic. The group of patients with no significant change in their genomic structure had relatively shorter time to progression (median = 5.5 months; range: 3–8 months) signifying the presence of already transformed cells with slow impact on clinical manifestations, which are currently driving the definition of the disease. In our small series, the pattern of genomic progression or its timing did not correlate with presence or absence of conventional, cytogenetic-defined, high-risk features.

The rearrangement profile could change during evolution in terms of absolute numbers (Fig. 3a), with a fraction of cases

showing prominent loss of some and gain of other events (Supplementary Figure 3). IGH translocations were, however, always found in the ancestral clone and as such remained stable during progression, confirming they are present in the first transformed cell and from there in all cells of the tumor (Fig. 3b). On the other hand, MYC rearrangements where mostly subclonal in the first sample and showed a global increase of their CCF at progression (Fig. 3b), confirming the view of such translocation as late events in MM evolution[3]. However, one patient surprisingly showed a decrease in cells bearing translocated MYC at progression (Fig. 3b), challenging the universal driver role usually attributed to this event. Furthermore, several nonrecurrent rearrangements barely detectable at the smoldering stage became clonal at the time of progression (Fig. 3c). This confirms that the proliferative advantage needed for tumor progression can be conferred by secondary rearrangements, and the identification of the driver role of each will be a challenge for future studies. Overall, we found a good concordance of the two evolutionary patterns in terms of changes in point mutations, CNAs and rearrangements, suggesting that the mutational processes acting at these three levels were broadly in concert over time and across subclones during evolution.

**Mutational signatures in smoldering myeloma.** Each genomic event, be it a point mutation, an indel, or a rearrangement, results from a specific mutational process that may drive the initial clonal expansion and subsequent evolution. Each process preferentially induces a certain nucleotide change within a certain 5′ and 3′ context, which is identified as a specific "signature".

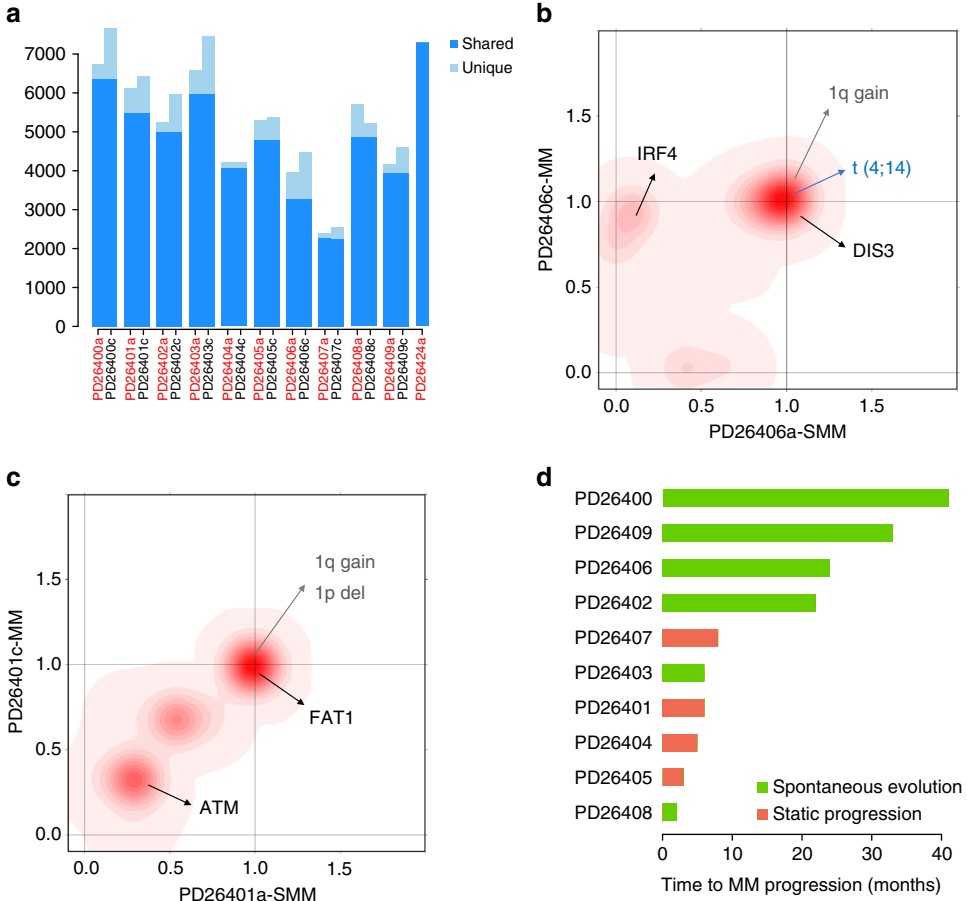

**Fig. 2** The genomic landscape of smoldering MM progression into symptomatic MM. **a** Comparison of substituion burden between SMM and MM, where dark blue represents shared mutations, and light blue mutations private to either sample. **b** Two-dimensional density plots showing the clustering of the fraction of tumor cells carrying each mutation at each time point; on x-axis and y-axis are plotted the SMM and MM phase, respectively. Increasing intensity of red indicates the location of a high-posterior probability of a cluster. In this case (PD26406), we have three main clusters, one was shared by 100% of cells both in SMM and MM; one was present only during the smoldering phase, and one appeared during progression. This is a typical example of spontaneous evolution model. **c** In this sample, no significant changes occurred in all main three clusters at progression. This is a typical example of the static progression model. **d** Time to progression of each case, where spontaneous evolution cases are in green, and static progression cases are in orange

Considering 6 possible substitutions in pyrimidine context, and 4 possible bases each at the neighboring 5′ and 3′ positions, there are then 96 possible combinations of substitutions in a trinu-cleotide context. We employed a nonnegative matrix factorization (NNMF) and model selection approach to extract mutational signatures[23,33] from the 96-class profile of the entire cohort (Fig. 4a) identifying 5 main clusters covering the majority of the mutational repertoire (Fig. 4b). Specifically, age-related (#1 and #5 of the original paper from Alexandrov et al.[23]) and APOBEC signatures (#2 and #13) accounted for 23% (3.2–40%) and 13% (range: 1–21%) of all substitutions, respectively (Fig. 4a)[23,33]. Furthermore, we found two additional signatures so far not implicated in myeloma: the noncanonical AID (Signature #9), contributing to 28% of all substitutions (range: 17–55%), and a fourth compatible with signature #8, accounting for 28% of all substitutions (range: 13–45%), previously described in different cancers and pertaining to a yet unknown mutational process (Fig. 4b)[23,29]. Finally, the fifth signature extracted by NNMF did not match any of the ones previously described, representing a potential novel and specific MM mutational process that we have defined as MM-1 (present in 7% of variants, range: 1–16%), whose pathogenesis remains entirely unknown at present. The same 5 clusters were validated at comparable frequencies in an already published, independent WGS series of MM at diagnosis or

relapse (Supplementary Figure 7, clinical information in ref. [12]). The prevalence of each signature varied between patients both in absolute and relative contribution, confirming genomic complexity and heterogeneity of MM (Fig. 4c, d). The nc-AID contribution was more prevalent among noncoding regions (Fig. 4e, f) than in coding regions, explaining why it was not found in prior WES studies[11,25]. Together, these data highlight that all processes that shape the MM genome are already operative at asymptomatic stages.

**Localized hypermutation in smoldering myeloma.** Somatic mutations were not evenly scattered across the genome. Instead, we found areas of localized hypermutation, termed "kataegis", which were occasionally reported in MM exomes[11]. In our whole genome data, we extend these observations and describe extensive evidence of kataegis, of which we defined 127 regions with a median of 6 (range: 3–17) per sample (Fig. 5a). Some areas of kataegis were expected, i.e., in the immunoglobulin heavy (IGH) or light (IGK/IGL) chain loci secondary to the physiological process of somatic hypermutation during B-cell development in the germinal center. IGH/IGK/IGL kataegis was found in all patients. After excluding these regions, ≥1 kataegis event was detected in 7/11 patients, for a total of 45 events.

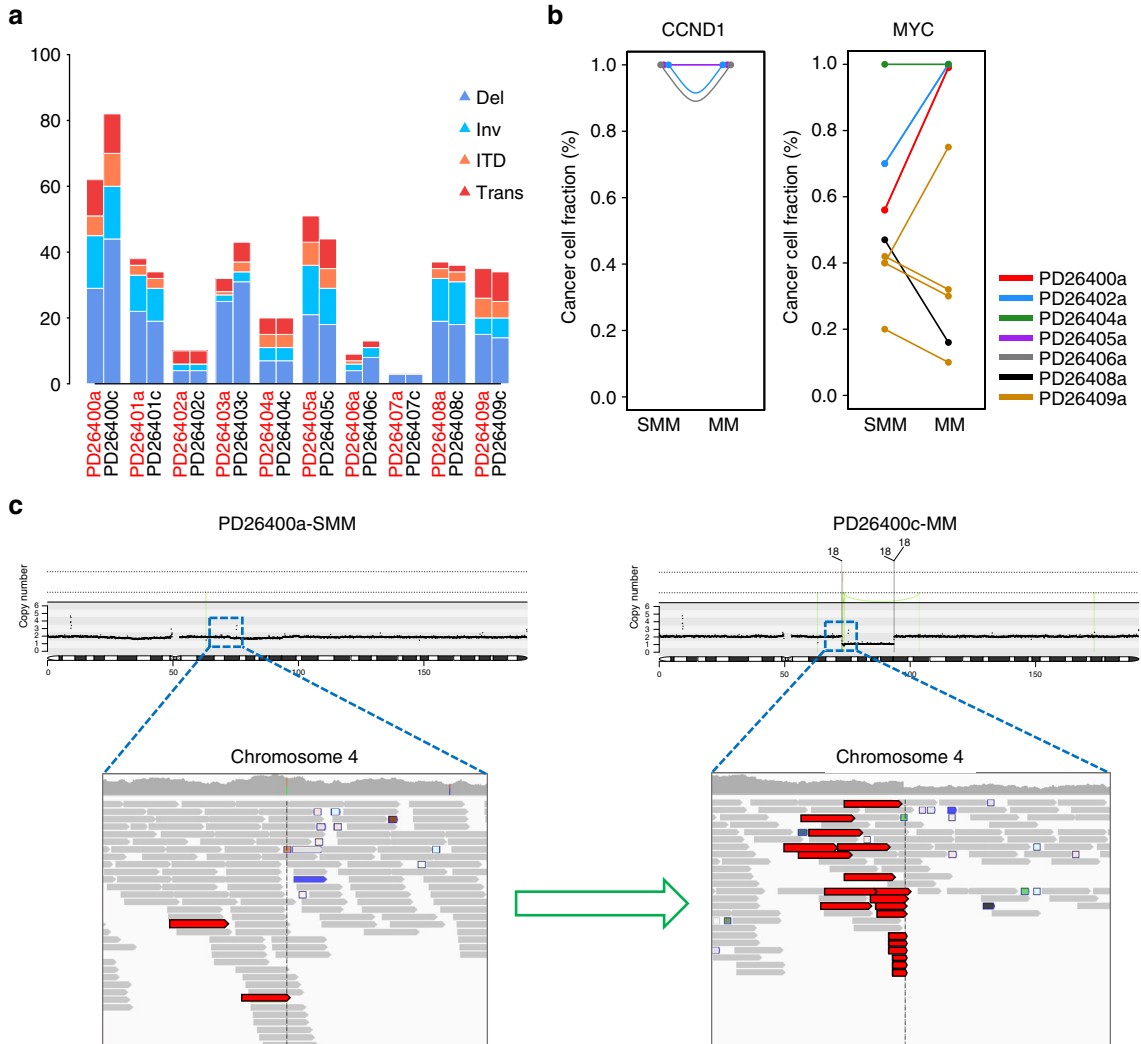

**Fig. 3** Rearrangements prevalence and involvement in smoldering MM progression. **a** Bar plot representing the rearrangement prevalence in all smoldering (x-label in red) and symptomatic MM patients (x-label in black), broken down by rearrangement type. **b** IGH (left) and MYC (right) translocated cases are plotted by allelic fraction changes during progression. Each line is color-coded for each patient. Notably patient PD26409 (yellow) had four independent MYC rearrangements, but only one increased its clonality upon progression to MM. **c** An example of progression associated with evolution of the clonal fraction of a translocation with unknown driver potential

Only two non-IGH/IGK/IGL kataegis events within the same patient were not conserved during SMM progression, suggesting most represent early events. Interestingly, kataegis was associated with rearrangements, found in 60% of such regions involving non-IGH/IGK/IGL loci. Breakpoints of such rearrangements were significantly closer to the kataegis region than expected by chance (Fig. 5b, **top**), suggesting they may arise as part of the same mutational process. In immunoglobulin regions, where rearrangements were mostly composed of deletions from the V(D)J recombination and class switch recombination processes, this phenomenon was even more pronounced (Fig. 5b, **bottom**). To look for mutational processes operative around these events, we restricted mutational signature analysis to kataegis regions, where we extracted 3 main signatures: nc-AID, APOBEC, and a third process not included in COSMIC mutational signatures data set (http://cancer.sanger.ac.uk/cosmic/signatures). The profile of this latter process was most similar to the canonical AID (c-AID) mutational signature recently described in a chronic lymphocytic leukemia WGS study (Fig. 5c)[29], and it was likely missed by previous NNMF studies because it is very localized in the

genome and present in few types of cancers only. Consistent with c-AID activity, we found this signature to be particularly prevalent in IGH/IGK/IGL loci (Fig. 5d), and less so in other regions (Fig. 5e). Overall, the combined effect of c-AID and nc-AID was responsible for more than 70% of all substitutions in kataegis regions (Fig. 5d, e), suggesting a causative role of aberrant AID activity in shaping the early mutational repertoire of neoplastic plasma cells, and not just a legacy of its physiological activity in the germinal center.

**Evolution of signatures over time**. Comparing paired samples, no significant differences were observed in the prevalence of the five signatures during evolution, both in terms of absolute numbers and relative contribution (Fig. 4d, e). These data suggest that, independent from patterns of genomic evolution, the mutational processes that shape the MM cell genome are already operative at the smoldering stage. However, as mutations could be clustered into clonal (early, present in the first transformed cell and from there in all cells of the tumor) or subclonal (late, acquired by a fraction of tumor cells after transformation and

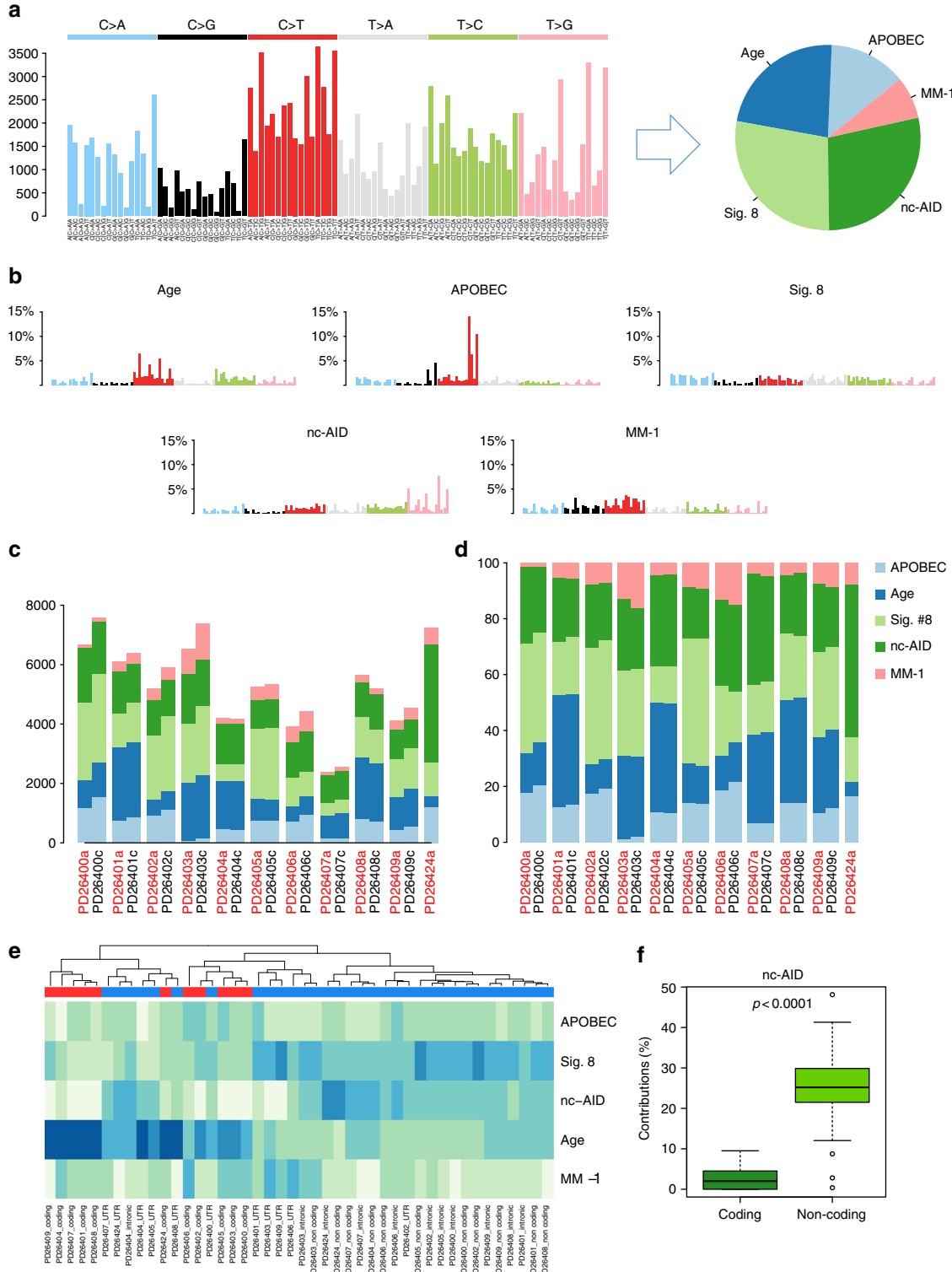

**Fig. 4** The landscape of mutational signatures involved in MM. **a** The 96-substitution class prevalence in all samples in the study from which NNMF extracted five main processes. **b** Representation of the five processes extracted by NNMF. **c**, **d** Barplots representing the absolute (**c**) and the relative (**d**) contribution of each mutational signature for each sample. **e** Hierarchical clustering based on the relative contribution of each mutational signature in each patient, according to the coding or noncoding status of each mutation. **f** Boxplot showing a strong association between nc-AID process and noncoding mutations. The p value was extracted by Wilcoxon test (wilcox.test R function). The whiskers are proportional to the interquartile range and are plotted with default parameters using the boxplot.stats R function

closer to the sampling time), we asked whether the contribution of the five signatures varied in the preclinical phases of MM development, i.e., before the actual sampling at the SMM stage. When each cluster of mutations was analyzed as an independent

genome, NNMF reported striking differences in prevalence of specific mutational signatures between clonal and subclonal events (Fig. 6a–c and Supplementary Figure 8). Specifically, nc-AID contributed to 47% (range: 36–59) of all early substitutions

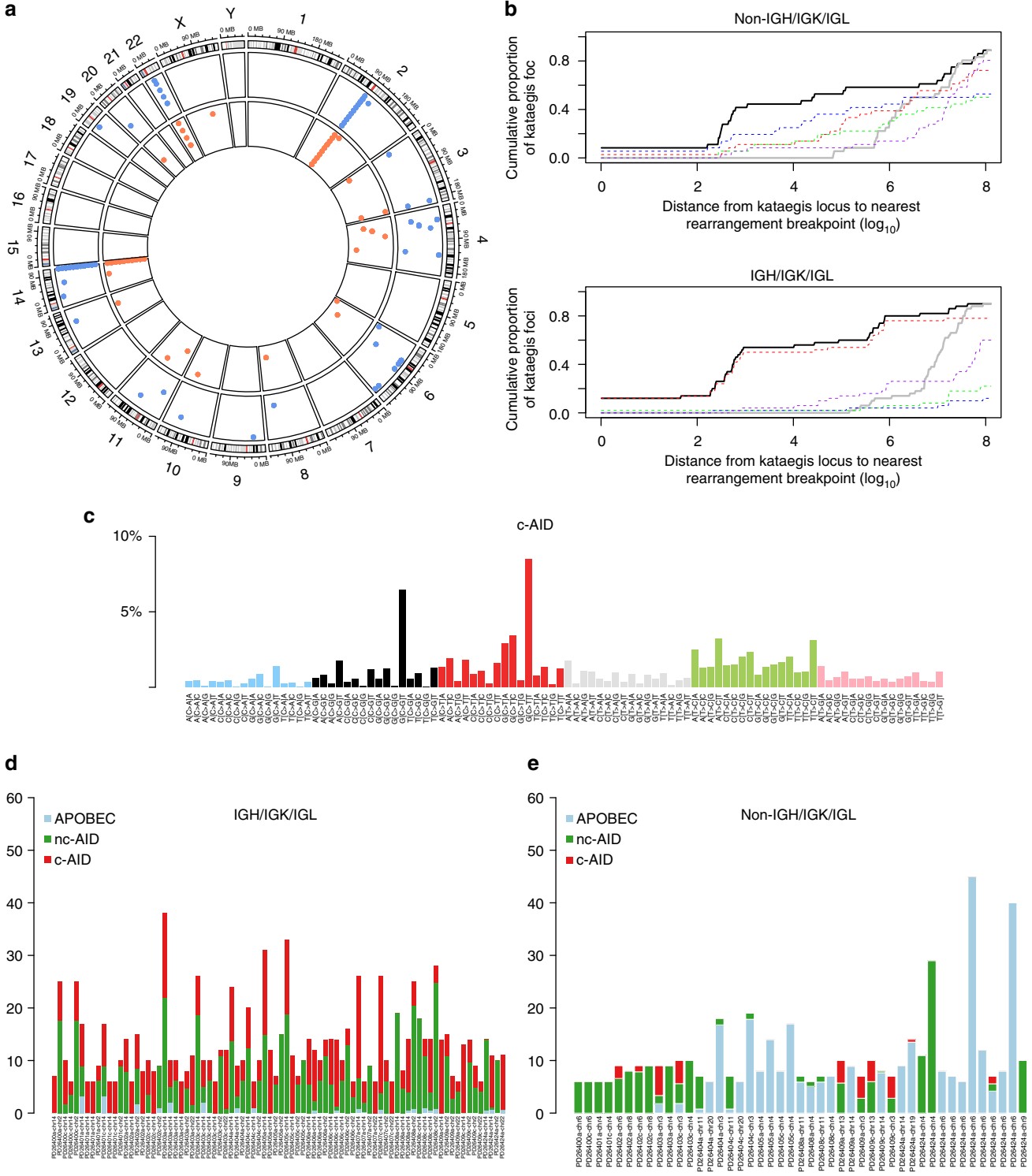

**Fig. 5** The prevalence, role and origin of localized hypermutation in MM. **a** Circos plot representing the distribution of all kataegis event among smoldering MM (blue dots) and symptomatic MM (orange dot). **b** Kataegis events were frequently found near rearrangements (black line representing the actual distance from the breakpoint). This association was higher than expected by chance (gray line). The rearrangements are broken down by type (blue line = inversion, red line = deletions, green line = ITD, and purple line = translocation). **c** The 96-mutational class histogram of the canonical AID signature extracted by NNMF on localized hypermutation events. **d**, **e** The absolute contribution of each involved mutational signature among all localized hypermutation regions within the IGH/IGK/IGL (**d**) and the non-IGH/IGK/IGL (**e**)

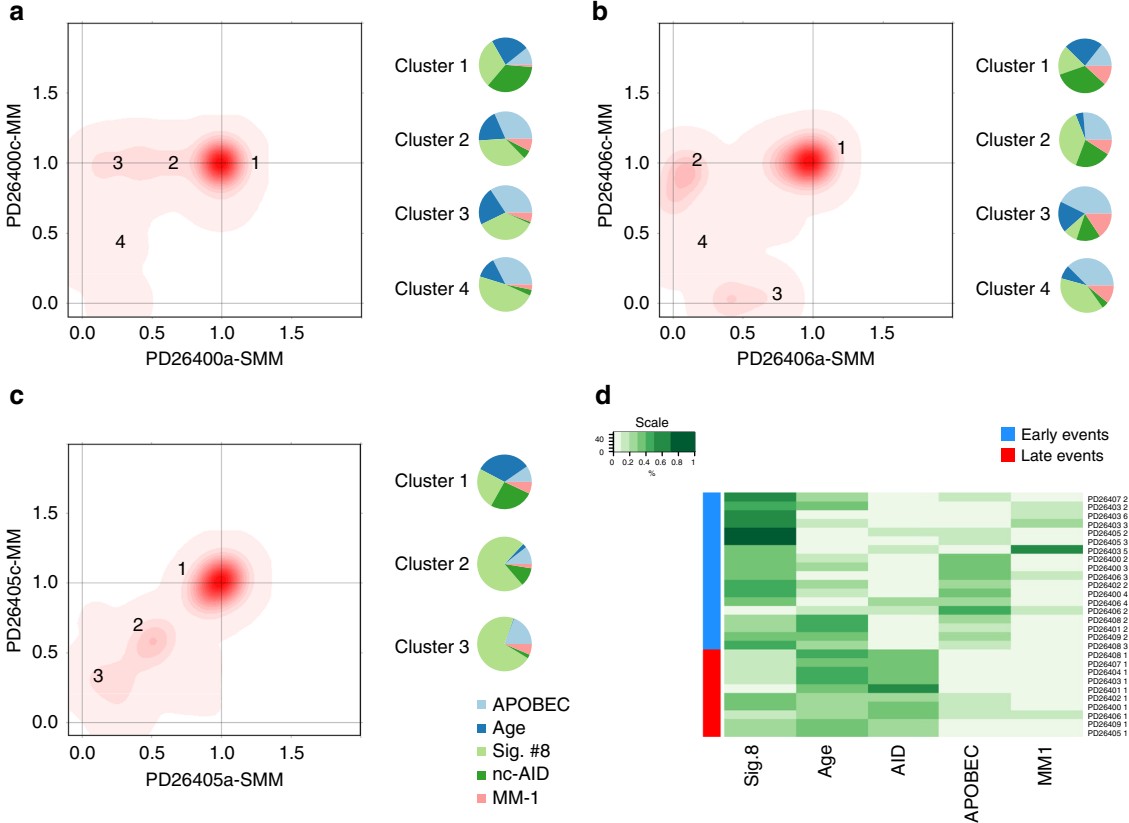

**Fig. 6** Mutational signature contribution during SMM progression. **a–c** Three examples of different mutational signatures contribution during SMM progression. The nc-AID contribution was particularly enriched in cluster 1 (clonal events) of all patients. Conversely, its contribution was significantly decreased or virtually absent among all later events. This different chronological mutational signature activity was observed in all investigated cases, where all early clonal events were clustered together in a cluster enriched for nc-AID activity (**d**)

in all patients. Conversely, its activity was minimal, if not absent, among late substitutions, where instead APOBEC and Signature #8 were responsible for more than 50% of all substitutions in all patients [27% (range: 3–58) and 18% (range: 7–50), respectively]. Unsupervised hierarchical clustering confirmed all early clones had a similar signature contribution that was different from late clones (Fig. 6d).

## Discussion

In our characterization of the genomic landscape of SMM we observe that cytogenetic, mutational, and rearrangement profiles are very similar to what has been described in MM. The most common aberrations (gain 1q, del13q, hyperdiploidy, and IGH translocations) were all clonal and therefore retained in MM, underlying their role in early stages of the disease. Interestingly, all SMM samples analyzed in this study were characterized by a significant subclonal heterogeneity that was frequently perturbed at the time of progression. While somewhat counterintuitive, the complexity of SMM genomic profile did not seem to be associated with a shorter time to progression nor to a more frequent evolution through accumulation of additional changes. PD26400 represents an emblematic example of this, being the patient with the longest time to progression yet the most complex genomic profile (Supplementary Figure 3). Likely, these differences reflect a highly variable mutation rate between patients that does not correlate with clinical aggressiveness.

Using paired samples, we report two very important and translationally significant results. First, we observe two models of progression from premalignant stage to MM. In the "static

progression model", the same subclonal architecture was retained as the disease progressed to MM. In this model, the time to progression solely reflected the time needed to accumulate a sufficient disease burden to become clinically symptomatic. Here, all the genomic features of an overt myeloma were already present when the disease was defined as SMM based on clinical parameters. Moving forward the task will be to genomically redefine symptomatic myeloma so as to include these SMM patients in the category that should be treated as MM. An early identification and treatment of these patients will be the direction of future studies. The "spontaneous evolution model" represents an interesting example of spontaneous Darwinian evolution where the subclonal composition of smoldering MM changed without any selective pressure from treatment, owing to the stochastic acquisition of additional mutations conferring a proliferative advantage to one of the subclones. While in the first model the time to progression is generally lower than 1 year, in the case of spontaneous evolution the median time to progression was longer and reflected the time needed for the generation of a truly new malignant clone that would progress to overt MM. Patients in this group will be the candidates to undergo preventive therapeutic strategies to truly abrogate progression to myeloma.

A second important finding identifies processes operative at the early premalignant stage and then later related with progression to myeloma. Our study shows that all samples (SMM and MM) are characterized by an early and major contribution from AID to the generation of the mutational spectrum of the transformed postgerminal center B-cell giving rise to the myeloma clone. This distinct profile is shared by all patients and represents an early

common driver mutational process, consistent with AID activity in the germinal center and its absence in MM cells[34]. Based on these data, we hypothesize a novel pathogenic model for MM where aberrant AID activity contributes to tumor initiation, and provides a fertile ground where other later processes (i.e., APOBEC and signature #8) act and shape the final genomic landscape of overt MM. Consistently, we have recently shown that APOBEC activity increases with disease progression from MGUS to plasma cell leukemia, reinforcing this finding[16].

Compared to previous WES studies, our WGS provided a much richer catalog of genetic lesions but a shallower median coverage. As a result, potentially we may have missed additional subclones present at low frequencies in the cancer samples, overestimating the percentage of cases where no change in the genomic structure occurred. As sequencing costs will fall, future studies may take advantage of the wealth of data provided by WGS at the depth of WES studies, and add sensitivity to our initial observation. Our data raise potential caveats for the search of markers of progression. In fact, current techniques will mostly allow identification of genomic events affecting the main SMM clone only, but this is not necessarily the one that will progress. In our series in fact, most samples showing subclonal changes acquired novel mutations not evident at the time of the first sampling. Again, deeper coverage will be required to add sensitivity and bring advances that may be clinically useful.

In summary, our unique paired samples from SMM and symptomatic MM from the same individual provide an important insight into both patterns of progression and mechanism driving the biological processes related with genomic evolution at different stages of the disease. These results also highlight potential future translational approaches and point to redefining both smoldering as well as symptomatic myeloma.

## Methods

**Sample selection.** The study involved the use of human samples, which were collected after written informed consent was obtained. All patients were diagnosed between August 2008 and January 2014 before the revised International Myeloma Working Group diagnostic criteria were published[5]. Based on the new criteria, patient PD26424 would have been classified as active MM because of 61% BM plasma cells (Supplementary Table 1). However, this patient was considered as SMM based on prevailing criteria at the time and was included in this study. As this patient received treatment prior to obtaining a progression sample, it was not included in the analysis of progression.

The protocol was approved by RES Committee East of England—Cambridge Central (WTSI protocol number 15/046). Samples and data were obtained after an informed consent was signed, and managed in accordance with the Declaration of Helsinki.

DNA were extracted from 21 samples from CD138+ myeloma cells purified from bone marrow, and constitutional control DNA originated from peripheral blood mononuclear cells. Purity of the CD138+ fraction was assessed by anti-CD138 immunocytochemistry post sorting, and only samples with >90% plasma cells were sequenced. For 10 patients, we sequenced 2 different samples collected at different time points of the disease.

**Massively parallel sequencing and alignment.** Short insert—500 bp—genomic libraries were constructed, flowcells prepared and sequencing clusters generated according to Illumina protocols. We performed 100 bp paired-end sequencing on HiSeq ×10 genome analyzers. The average sequence coverage was 38.7-fold. Short insert paired-end reads were aligned to the reference human genome (GRCh37) using Burrows–Wheeler Aligner, BWA (v0.5.9)[35].

**Processing of genomic data.** CaVEMan (Cancer Variants Through Expectation Maximization: http://cancerit.github.io/CaVEMan/) was used to call somatic substitutions[36]. Indels were called using a modified Pindel version 2.0. (http://cancerit.github.io/cgpPindel/) on the NCBI37 genome build[37]. Structural variants were discovered using a bespoke algorithm, BRASS (BReakpoint AnalySiS) (https://github.com/cancerit/BRASS) through discordantly mapping paired-end reads[38]. Discordantly mapping read pairs that were likely to span breakpoints, as well as a selection of nearby properly paired reads, were grouped for each region of interest. All rearrangements identified by BRASS that failed high-quality mapping of split reads were manually evaluated by IGV. Considering the number of reads supporting each rearrangement breakpoints and adjusting this value for both copy

number and ACF, we were able to estimate the adjusted variant allelic frequency (VAF) of each rearrangement VAF. This parameter was used to roughly estimate the change in the number of cells carrying each rearrangement during progression.

Allele-specific copy number analysis of tumors was performed applying ASCAT (v2.1.1) on NGS data[39,40]. The evaluation of copy number changes for the identification of subclonal aberrations was also performed by Battenberg as previously described[41].

To model clusters of clonal and subclonal point mutations, allowing inference of the number of subclones and the fraction of cells within each subclone, we used a 2-D Bayesian Dirichlet analysis as previously described for all patients with paired samples[11,42]. We classified as "spontaneous evolution", all progressions that matched with the previously defined clonal evolution models ("Linear", "Differential Clonal Response", and "Branching Evolution")[11], i.e., where the cancer cell fraction of the various subclones changed significantly between the two samples; conversely "static progression" was used to classify the ones that progressed without any significant genomic change.

**Mutational processes and signature analysis.** Signatures of mutational processes were analyzed using the Wellcome Trust Sanger Institute mutational signatures framework (NNMF)[23,33]. Because NNMF works best when a large number of samples is used[33], we increased our cohort size by adding 19 additional MM samples (4 newly diagnosed and 15 relapse), sequenced and analyzed by WGS at our institution with the same pipeline as described above (Supplementary Table 4). Our final cohort for NNMF thus included a total of 40 samples. The signature analysis of each subclone, represented by a cluster of mutations identified by the Dirichlet process, was limited to clusters with more than 100 substitutions.

Wilcoxon test (wilcox.test R function) was used to investigate the different mutational load and signature contribution between SMM and MM. All other analyses were performed using appropriate functions in R 3.3.2 software – (www.r-project.org). Code is available upon request.

**Data availability.** Sequence files are available at the European Genome-phenome archive under the Accession code EGAD00001001898.

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

## Acknowledgments

N.B. is funded by AIRC (Associazione Italiana per la Ricerca sul Cancro) through a MFAG (n.17658). F.M. is supported by AIL (Associazione Italiana Contro le Leucemie-Linfomi e Mieloma ONLUS) and by SIES (Società Italiana di Ematologia Sperimentale). This work was supported by Department of Veterans Affairs Merit Review Award I01BX001584-01 (NCM), NIH grants P01-155258 (NCM, HAL, MF, PC, and KCA) and 5P50 CA100707-13 (NCM, HAL, and KCA) and Leukemia and Lymphoma Society translational research grant (NCM).

## Author Contributions

N.M., P.C., and H.A.V. designed the study, collected, and analyzed the data and wrote the paper; N.B. and F.Mau. collected and analyzed the data and wrote the paper; S.M. designed the study and collected data; M.P., R.S., M.K.S., M.A.S., M.F., Y.T.T., F.Mag., P. M., P.Co., K.A. collected the data; D.G., A.F., I.M., K.J.D., J.Z., P.T., H.D., L.A.m D.C.W. analyzed the data.

## Additional information

**Competing interests:** The authors declare no competing interests.

