## [Peer Review File · Nature Communications]

REVIEWERS' COMMENTS:

Reviewer #1 (Remarks to the Author):

Aside from issues already raised which in many ways have been simply batted away the real issue relates to the depth of sequencing performed. TO me its never going to address the CNVs well. Hence with a coverage of ~30x it is likely that it is very difficult to make any solid statements about these regions, in particular NMF analysis on sub-clonal vs clonal changes etc.

AS previously stated its basically a sequencing project with no associated omics to inform biology.

Reviewer #2 (Remarks to the Author):

The manuscript investigates the genomic pattern of evolution from indolent (MGUS, SMM) to symptomatic stages of multiple myeloma (MM) using whole genome sequencing (WGS) of a unique set of paired patient samples. The manuscript is of high technical quality and adds substantial novel information regarding the genomic progression of early stages of myeloma. In the revised version of the manuscript all my concerns have been sufficiently addressed by the authors. The text has been altered accordingly- in particular, the discussion has been amended to better explain the correlation between mutational signatures and disease progression.

Reviewer #3 (Remarks to the Author):

The authors have satisfactorily responded to the majority of comments from my prior review.

A minor comment remaining is that the composition of the independent WGS dataset used for validation of the 5 clusters, n=25 MM cases (Supplementary Fig 7) still does not appear to be provided. A brief description, as to whether these were newly diagnosed MM, relapse, etc. should be included.

Reviewer #1:

1. **Aside from issues already raised which in many ways have been simply batted away the real issue relates to the depth of sequencing performed. TO me its never going to address the CNVs well. Hence with a coverage of ~30x it is likely that it is very difficult to make any solid statements about these regions, in particular NNMF analysis on sub-clonal vs clonal changes etc. AS previously stated its basically a sequencing project with no associated omics to inform biology.**

As we have stated previously and Reviewers 2 and 3 have also commented, this is a unique data which for the first time investigates genomic changes in paired samples from SMM progressing to MM. We would also like to reemphasize that the WGS data analysis to call copy number events and subclonality has been validated by several groups including ours. We do accept its limitations in defining various levels of subclonality, which is not the focus of this paper at all.

Reviewer #2:

2. **The manuscript investigates the genomic pattern of evolution from indolent (MGUS, SMM) to symptomatic stages of multiple myeloma (MM) using whole genome sequencing (WGS) of a unique set of paired patient samples. The manuscript is of high technical quality and adds substantial novel information regarding the genomic progression of early stages of myeloma. In the revised version of the manuscript all my concerns have been sufficiently addressed by the authors. The text has been altered accordingly- in particular, the discussion has been amended to better explain the correlation between mutational signatures and disease progression.**

We appreciate the reviewer's comments.

Reviewer #3

3. **The authors have satisfactorily responded to the majority of comments from my prior review. A minor comment remaining is that the composition of the independent WGS dataset used for validation of the 5 clusters, n=25 MM cases (Supplementary Fig 7) still does not appear to be provided. A brief description, as to whether these were newly diagnosed MM, relapse, etc. should be included.**

We agree with the reviewer that clinical status of the samples was not stated clearly enough in the original version of the text. Detailed sample information is available as supplementary information of the original publication, referenced in our manuscript. In the revised version of our manuscript, we have added information on the diagnosis-relapse status for these samples in the supplementary figure 7.